# Estimating intra- and inter-subject oxygen consumption in outdoor human gait using multiple neural network approaches

Philipp Müller[1]*, Khoa Pham-Dinh[1], Huy Trinh[1], Anton Rauhameri[2], Neil J. Cronin[3]

**1** Faculty of Information Technology and Communication Sciences, Tampere University, Tampere, Finland, **2** Faculty of Medicine and Health Sciences, Tampere University, Tampere, Finland, **3** Faculty of Sport and Health Sciences, University of Jyväskylä, Jyväskylä, Finland

☯ These authors contributed equally to this work.
\* philipp.muller@tuni.fi

**Data Availability Statement:** Our dataset is available for download at https://doi.org/10.23729/a050e440-6f41-498d-8a31-097ff6881544.

## Abstract

Oxygen consumption ($\dot{V}O_2$) is an important measure for exercise test, such as walking and running, that can be measured outdoors using portable spirometers or metabolic analyzers. However, these devices are not feasible for regular use by consumers as they intervene with the user's physical integrity, and are expensive and difficult to operate. To circumvent these drawbacks, indirect estimation of $\dot{V}O_2$ using neural networks combined with motion features and heart rate measurements collected with consumer-grade sensors has been shown to yield reasonably accurate $\dot{V}O_2$ for intra-subject estimation. However, estimating $\dot{V}O_2$ with neural networks trained with data from other individuals than the user, known as inter-subject estimation, remains an open problem. In this paper, five types of neural network architectures were tested in various configurations for inter-subject $\dot{V}O_2$ estimation. To analyse predictive performance, data from 16 participants walking and running at speeds between 1.0 m/s and 3.3 m/s were used. The most promising approach was Xception network, which yielded average estimation errors as low as 2.43 ml×min$^{-1}$×kg$^{-1}$, suggesting that it could be used by athletes and running enthusiasts for monitoring their oxygen consumption over time to detect changes in their movement economy.

## Introduction

Oxygen consumption ($\dot{V}O_2$), also known as oxygen uptake, is frequently used to measure walking and running economy since the exchange of oxygen and carbon dioxide is highly correlated to energy metabolism. By monitoring $\dot{V}O_2$ over time, changes in movement economy due to training, rehabilitation, etc. can be detected. For unconstrained walking or running, in outdoor environments rather than on treadmills, $\dot{V}O_2$ can be measured directly by metabolic analyzers or portable spirometers, but these devices are inconvenient to use regularly, often require trained personnel for operation, and are expensive.

**Funding:** All authors received funding (as team members of a research consortium) from the Academy of Finland (https://www.aka.fi), grants 287295 and 323472. The funder had no role in the study design, data collection and analysis, decision to publish, or preparation of the manuscript.

**Competing interests:** The authors have declared that no competing interests exist.

Therefore, research on indirect estimation of $\dot{V}O_2$ from observations of surrogate features has received a lot of attention over the last two decades. Indirect estimation has benefitted from recent advances in machine learning techniques and development of consumer-grade, small, wearable sensors. The most common approach for $\dot{V}O_2$ estimation is based on Heart Rate (HR) measurements [1]. For example, several commercial products, such as the Suunto personal HR monitoring system, use HR data for estimating $\dot{V}O_2$ and energy expenditure [2]. Because HR is affected by age, sex, fitness level, exercise modality, environmental conditions, and day-to-day variability [3], HR index (HRI) is frequently used instead of HR [1, 4]. HRI is obtained by dividing the HR measurement by an individual's resting HR, which has the potential to remove the need for individual calibration [4]. In [2] additional features such as R-wave-to-R-wave (R-R) heartbeat intervals, R-R-based respiration rate, and on-and-off $\dot{V}O_2$ dynamics at various exercise conditions were used for $\dot{V}O_2$ estimation. However, authors of the study acknowledged the limitations in the estimation accuracy when including individual maximal $\dot{V}O_2$ and HR values. Several studies have used linear regression models to estimate $\dot{V}O_2$ [1–4], which worked well for moderate intensity exercises. However, for very low and very high intensity exercises the relationship between $\dot{V}O_2$ and HR is significantly nonlinear, resulting in poor $\dot{V}O_2$ estimates.

Other factors that can affect the relationship between $\dot{V}O_2$ and HR include altitude, exercise duration, hydration status, medication, state of training, and time of day [2 from pavel]. To account for these factors, in cycling breathing frequency, mechanical power, and pedaling cadence, which can be measured directly from cycling ergometers, can be included for $\dot{V}O_2$ estimation [5–7]. For walking or running in unconstrained outdoor environments, breathing frequency, cadence, speed, and speed variation calculated by wearable devices can be used as input features [3].

In [8] we computed motion features, namely step-wise average speed, peak-to-peak speed difference, step duration, and peak-to-peak difference in vertical movement from measurements of an inertial navigation system combined with a Global Positioning System (INS/GPS) device. The wearable INS/GPS device measured acceleration, velocity, angular velocity and orientation of the upper body (for details the reader is referred to [3]. The four motion features were used together with HR as input features for estimation of $\dot{V}O_2$ during walking and running by a long short-term memory (LSTM) neural network. The results suggest that LSTM neural networks are able to accurately estimate oxygen consumption; the achieved accuracy was 2.49 ml×min$^{-1}$×kg$^{-1}$ (95% limits of agreement). For comparison, in [9] $\dot{V}O_2$ during walking and other daily activities were estimated by random forest regression using breathing frequency, HR, hip acceleration, minute ventilation, and walking cadence; the achieved accuracy was 6.17 ml×min$^{-1}$×kg$^{-1}$ (95% limits of agreement).

One limitation of the study in [8] was that data for training the LSTM neural networks and for evaluating its performance came from the same individual (intra-subject estimation). However, estimating oxygen consumption for an individual by a model trained with data from the same individual is time-consuming and not always feasible. Ideally, the model would already be trained beforehand, using training data from other individuals, and used immediately for $\dot{V}O_2$ estimation. This is referred to as inter-subject estimation. Furthermore, in [8] only light and moderate intensity exercises were covered.

Therefore, in this paper a wide selection of neural network models are studied for estimating inter-subject oxygen consumption across a range of walking and running speeds (1.0 m/s to 3.3 m/s) on a level outdoor track based on measurements of motion features and heart rate. The contributions of our paper are three-fold. First, we demonstrate that by using an early exit

strategy and optimizing hyperparameters the accuracy of the LSTM model from [8] can be significantly improved (average estimation error was reduced by approximately 82%). Second, we show that with more sophisticated neural network structures $\dot{V}O_2$ estimates for inter-subject estimations can be obtained that are more accurate than the intra-subject estimations yielded by the LSTM model from [8]. Finally, a more detailed correlation analysis between the neural networks' input features (motion and HR data) and output feature ($\dot{V}O_2$) than in [8] is provided, which yields insights into why neural networks are able to yield accurate $\dot{V}O_2$ estimates.

## Materials and methods

### Experimental data

Sixteen healthy participants between 18 and 35 years of age (age 27.5 ± 3.5 yrs, height 175.3 ± 8.4 cm, body mass 71.8 ± 12.9 kg, body mass index 23.3 ± 3.4 kg/m$^2$, eight females) participated in the field tests on a level outdoor track. Ten of the participants were recreational runners, meaning that they ran at least twice per week during summer and performed other endurance sports during winter. Their statistics were: age 28.1 ± 3.7 yrs, height 177.6 ± 6.4 cm, body mass 70.8 ± 11.4 kg, body mass index 22.3 ± 2.7 kg/m$^2$, five females. For the remaining six participants age 26.5 ± 3.1 yrs, height 171.5 ± 10.59 cm, body mass 73.3 ± 16.1 kg, body mass index 24.8 ± 4.2 kg/m$^2$, three females. These participants ran at most twice per month. The Ethics Committee of the University of Jyväskylä approved the study. Participants were recruited between 5 May 2018 and 31 July 2018. All participants were informed about the content and purpose of the testing procedure, and provided written informed consent, witnessed by one researcher. The research was conducted in accordance with the World Medical Association Declaration of Helsinki [10].

Each participant was equipped with a datalogger, a portable spirometer, and a chest strap for measuring the heart rate. The datalogger was assembled on a Raspberry Pi 3 model B running Raspbian operating system, connecting with a high quality Vectornav VN-200 (Vectornav Technologies, United States) GPS-aided Inertial Navigation System (INS/GPS), a GPS antenna, and a battery. The inertial measurement unit incorporates an accelerometer, a gyroscope, a magnetometer and a barometric pressure sensor (details can be found in [3]). The device measures 150 x 75 x 48 mm and was carried on the participant's upper back in an orienteering battery vest. Oxygen consumption and other breathing parameters were measured during the walking, running and rest periods with a Jaeger Oxycon Mobile portable breath gas analyser (Viasys Healthcare GmbH, Germany). The setup consists of a desktop and a portable setup, with the latter including a sensorbox, a data exchange unit, and a mask to which a digital volume transducer and a gas tube were connected. Both sensorbox and data exchange unit were carried on the participant's upper back with a special vest so that the units were located on either side of the datalogger. Heart rate was measured using a Polar V800 heart rate monitor and an H10 strap with integrated heart rate sensor (Polar Electro Oy, Finland).

Participants were asked to rest for five minutes at the beginning of the measurement session to obtain oxygen consumption at rest, which enabled studying the effect of exercise on a participant's oxygen consumption. After that, participants were asked to walk or run along a 200 meter long track on the main straight of the level outdoor track at various speeds. Walking speeds were 1.0 m/s, 1.3 m/s, and 1.5 m/s; running speeds were 2.2 m/s, 2.5 m/s, 2.8 m/s, 3.1 m/s and 3.3 m/s. Subject 3, in addition, also ran at 3.6 m/s. Each participant started with walking at 1.0 m/s. The order of the remaining seven speeds were randomized for each participant individually by a browser-based randomizer (http://www.random.org/lists). Speed was controlled by LED modules spaced at one meter along the track, which enabled control of speed

with an accuracy of 0.1 m/s. Participants were asked to follow the lights while walking/running for five minutes for each speed. After each walking/running speed, participants stopped for a few seconds and then returned to the starting point to sit still for five minutes, allowing heart rate and oxygen consumption to return to resting levels.

The Oxycon Mobile spirometer measured every five seconds $\dot{V}O_2$ and respiratory frequency using breath-by-breath methods. To ensure accurate measurements, the spirometer was re-calibrated at the start of each measurement session.

In line with [8], oxygen consumption measurements were smoothed by applying a Savitzky-Golay filter [11] with polynomial order and window length set to 3 and 1 respectively three times. Due to unusually noisy data, for subject 7 also polynomial order set to 9 was tested and used in inter-subject estimations. Heart rate was recorded continuously (beat-by-beat) during the test at a sampling rate of 1 Hz. Data were smoothed and interpolated after the tests. Smoothing was done by a moving average with window length 3. For subjects 4 and 9 the window length was increased to 5 due to the exceptionally noisy heart rate data. The INS/GPS datalogger recorded acceleration, velocity, angular velocity and orientation at 400 Hz and saved them to a memory card through a wired connection, preventing any data loss. Accuracy levels of speed and speed difference were approximately 0.05 m/s; accuracies of computed vertical oscillation and step duration were about 1 cm and 10 ms respectively (for more details refer to [3]).

After the measurement campaign it was noticed that for subject 1 heart rate data was only partly available. Thus, only parts for which heart rate data as well as data from the INS/GPS datalogger and the portable spirometer were available were used for analysis. Similarly, the dataset for subject 2 lacked spirometer data for approximately 90 s, thus all data from this period was removed during data preprocessing. Subject 10 terminated the measurement campaign after the fifth walking/running cycle. Due to the limited number of participants, data from these five cycles were, nevertheless, included in the analysis.

## Dataset preparation

After data collection, measurements of spirometer, INS/GPS datalogger, and heart rate device were synchronized in time. Synchronization of spirometer and INS/GPS datalogger data was based on the internal clocks of both devices. For oxygen and heart rate time series data the cross-correlation of standardised oxygen and heart rate data was calculated and the highest peak was used as offset estimate for synchronization.

Step segmentation described in [3] was applied to motion data from the INS/GPS datalogger and used to compute walking/running metrics commonly used in gait analysis on a step-by-step basis (see [3] for details). Accelerations and velocities were computed in the anatomical frame. In the anatomical frame the x-axis is pointing into the direction of progression (anterior direction) and the z-axis is pointing upwards, parallel to the field of gravity. The y-axis is perpendicular to x- and z-axes and completes a right-handed coordinate system. Oxygen consumption and heart rate measurements were resampled to match the step-by-step frequency of walking/running metrics.

In [8] feature engineering was used to identify features derived from the INS/GPS data and heart rate data that yielded most accurate estimates for oxygen consumption, the so-called target feature, when being used as inputs for a long short-term memory neural network. The following five input features were validated in [8] based on consider-only-one and leave-one-out approaches, and were used here as well:

- Speed: arithmetic mean of the velocity (= step length/duration of step) over one step, measured in m/s

- Speed change: peak-to-peak difference in speed during one step, measured in m/s

- Step duration: measured in s

- Vertical oscillation: peak-to-peak difference in vertical movement, measured in m

- Heart rate: measured in bpm

Sequences of steps were used as inputs for the network training. The experiments for [8] indicated that input sequences of 50 steps yielded satisfactory estimates of the time-dependent decay between oxygen consumption (target feature) and past input values, thus the same length was used in this paper as starting point and inputs were 5-by-50 matrices, with each row containing the sequence of one of the aforementioned input features.

## LSTM network architecture for intra-subject estimations

In [8] a many-to-one long short-term memory (LSTM, [12]) model was developed and tested successfully for intra-subject oxygen consumption ($\dot{V}O_2$) estimation. The network structure was simple and consisted of a sequence input layer, a LSTM layer with 150 hidden units, one dense layer, and an output layer. For training the network the *Adam* optimizer was used; the learning rate was set to 0.005; and training was run over 8 000 epochs. The motivation for using such a large number of epochs was that only a small dataset was available (subjects only walked/ran four times three minutes at four different speeds). Since a more extensive dataset has been collected for this paper, the *LSTM model* from [8] was trained here for 1 000 epochs, because preliminary tests showed that training and validation loss converged within 1 000 epochs.

For [8] the aim was to demonstrate that LSTM networks can successfully estimate oxygen consumption, rather than finding the most accurate network model. For this paper several alternative network structures were tested. First, twelve simple modifications of the LSTM model were studied. The most promising modification was an Early Exit Neural Network (e.g. [13]) that used 100 instead of 150 hidden units in the LSTM layer. The initial learning rate of 0.005 was decreased by factor 0.2 if the validation loss did not improve over the last 25 epochs. The minimum learning rate was set to $10^{-6}$. Furthermore, in order to avoid overfitting, training was terminated if over the last 20 epochs no sequence of three epochs with decreasing difference between training and validation loss was observed. Hereafter, this model is referred to as *Modified LSTM model*.

The LSTM model [8] was compared with the Modified LSTM model by analysing their performances for intra-subject oxygen consumption estimation. Data from each of the 16 subjects was divided into training (70% of all samples), validation (15%), and test datasets (15%) randomly. The input sequences were normalised by removing the mean and scaling to unit variance to enable better fit and prevent divergence in the network training. After that for each subject both the LSTM model and the Modified LSTM model were trained. While the LSTM model was always trained for 1 000 epochs, Modified LSTM model was trained for 1 000 epochs or until the termination rule described above was fulfilled. The $\dot{V}O_2$ estimation capabilities of the both network structures were evaluated using the corresponding test datasets. This process was repeated for each subject five times (5-fold cross-validation) to assess how well the model generalises to different parts of the same participant's dataset.

## Network architecture for inter-subject estimations

For inter-subject estimations the simple neural networks used for intra-subject estimations are insufficient, since the physiological and biomechanical features of gait differ between

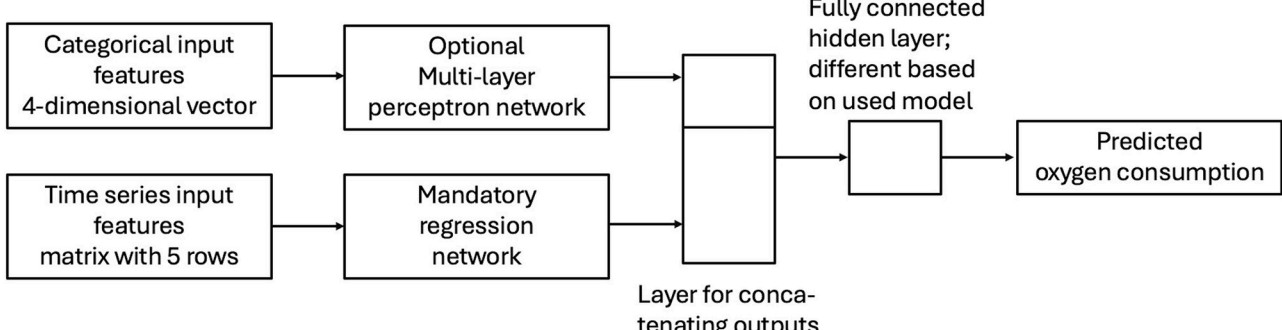

**Fig 1. Network architecture used for inter-subject $\dot{V}O_2$ estimation.** Upper, optional branch used four categorical variables age, body mass index, sex and whether or not the individual was a recreational runner as input data. Lower branch used temporal data for speed, speed change, step duration, vertical oscillation, and heart rate as input. Estimated value was $\dot{V}O_2$.

individuals. Therefore, in this paper a model architecture with a mandatory regression network and an optional feature network was used for a selection of more sophisticated neural network types (see Fig 1). This architecture allowed the model to process both temporal and categorical input data. Temporal data included the same features as for the networks meant for intra-subject estimation, namely speed, speed change, step duration, vertical oscillation, and heart rate. Categorical data consisted of four individual features of test subjects, which are described in Table 1.

The structure of the fully connected hidden layer in Fig 1 varied for the different tested network types. For recurrent neural networks and convolutional neural networks no hidden layer was used; for DenseNet the layer had output dimension 4, while for residual neural networks and Xception networks the output dimension was 16. In all three network types ReLU activation was used (more details are given in the sections below).

Each network type was furthermore trained in various configurations. Due to the relatively small size of the dataset 100 epochs proved to be sufficient for achieving convergence in the network training phase. For all networks the AdamW optimizer [14] and the cosine learning rate scheduler with an initial learning rate of $10^{-3}$ and a final learning rate of $10^{-5}$ were used. Batch size was set to 64.

For evaluating the network types and configurations for inter-subject $\dot{V}O_2$ estimation, leave-one-out cross-validation was used, which prevented data leakage (i.e. data from one subject being used for both training and testing). For each of the sixteen participants data from the remaining fifteen participants were used to train (data from thirteen participants) and validate (data from two participants) the networks and afterwards $\dot{V}O_2$ for the sixteenth subject

**Table 1. Overview of categorical features and their categories.**

| Feature | Category 0 | Category 1 | Category 2 |
|---|---|---|---|
| Age | $\leq$ 25 years | 26–29 years | $\geq$ 30 years |
| Body Mass Index (BMI) | <22 | 22–25 | >25 |
| Sex | female | male | - |
| Fitness level | untrained | trained | - |

Column *Feature* shows the name of the categorical input features. The remaining columns show the condition for being placed in one of the three categories. For features *Sex* and *Fitness level* only categories 0 and 1 were used.

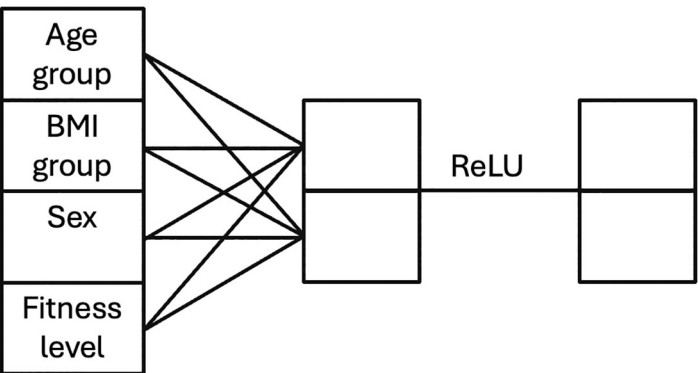

**Fig 2. Multi-layer perceptron for attributes vector with rectified linear unit (ReLU) activation function.** From left to right the MLP consists of a four-dimensional attributes vector, a two-dimensional vector, and a two-dimensional output vector.

was estimated with the trained models. This process was repeated five times for each of the sixteen participants (5-fold cross-validation). In the end the average root-mean square error (RMSE) and its corresponding standard deviation over all sixteen participants and five repetitions per participants were calculated.

**Multi-layer perceptron for processing participant-specific features.** The feature head is a multi-layer perceptron (MLP) with an output of two neurons. The network is described in Fig 2. A four-dimensional vector containing participant-specific features was used as input. It was fully connected to a two-dimensional output vector that was then forwarded to a ReLu function.

**Regression head.** The mandatory regression head, which used temporal data as input, was constructed with five different neural network types: conventional recurrent neural networks, convolutional neural networks, residual network, DenseNet, and Xception network. The output of every network type in the regression branch was a vector of size 16 or 32 that was used for regression or concatenation with the feature head.

*Recurrent neural networks.* Conventional recurrent neural networks (RNNs) can handle temporal data of any length using a recurrent hidden state that is updated at each time step using a nonlinear function that depends on the current measurement and the recurrent hidden state of the previous time step [15]. The long-term gradients of a conventional RNN being trained using back-propagation are at risk of converging to zero (so-called vanishing gradient problem), which effectively terminates the network to learn. Therefore, in this paper, besides recurrent neural network [16] layers, also long short-term memory [12], and gated recurrent units (see e.g. [15]) layers were tested. LSTM was developed to tackle the vanishing gradient problem. It is able to bridge long time intervals without sacrificing short time lag capabilities by "enforcing constant error flow through internal states of special units" [12]. These units act as a sort of memory and ensure that irrelevant information is forgotten and only important information is propagated. Gated recurrent units (GRUs) are able to capture dependencies of different time scales adaptively [15]. These units are similar to LSTM, but possess a gating mechanism for inserting or forgetting information and lack separate memory cells [15].

In this paper RNN configurations having three RNN, LSTM or GRU layers were studied. Each hidden layer contained 128 neurons. Both one-directional (see Fig 3(a)) and bi-directional versions (see Fig 3(b)) were tested.

*Convolutional neural networks.* In the experiment, a fully convolutional neural network (CNN), mentioned in [17], was also evaluated. The network contained three 1D-convolutional

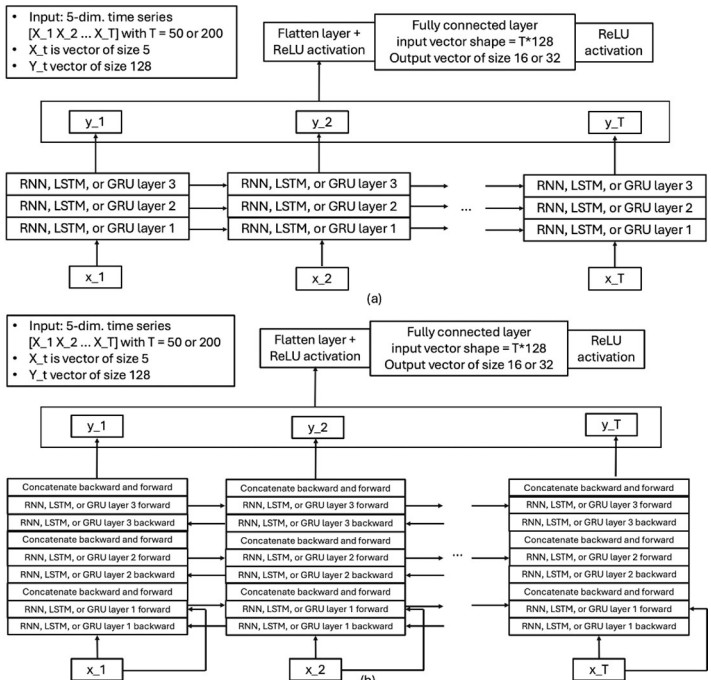

**Fig 3. Architectures of RNN-type networks.** One-directional versions are illustrated in (a), bi-directional versions in (b).

layers and a fully connected layer at the end. The output is a vector of size 16 or 32. The architecture of the CNN is displayed in Fig 4. Kernel size was set to three and the number of filters in the three convolutional layers were $n_f = \{32, 64, 32\}$. Two CNN configurations were implemented, one with 16 and one with 32 neurons in the last fully connected layer.

*Residual neural networks.* A residual network (ResNet) employs skip connections to facilitate the gradient flows during training. The architecture has been successfully applied to computer vision tasks [18], but also application on time series data have been studied (e.g. [17]). In this paper the architecture from [17] was used, with modifications to the hyperparameters to ensure that the model was compatible with the considerably smaller dataset.

The residual network was built from so-called residual blocks, which each contained three one-dimensional convolutional layers as well as a direct shortcut from input to output that used addition. In the experiment, three blocks with three convolutional layers each were used. For the kernel sizes $K_{ks}$ in each layer two different options were tested, $K_{ks} = \{3, 3, 3\}$ and $K_{ks} = \{7, 5, 3\}$. The number of filters in the three residual blocks were set to $\{n_f, 2n_f, 2n_f\}$ with $n_f = 24$. The dimension of the output vector ($c_{out}$) was set to 16. The third block was followed by a global average pooling layer and a fully connected layer. Fig 5 illustrates the tested network architecture.

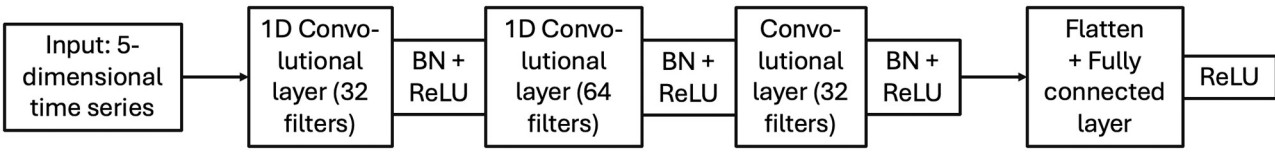

**Fig 4. Architecture of the convolutional neural network.** BN + ReLU stands for batch normalization followed by ReLU activation.

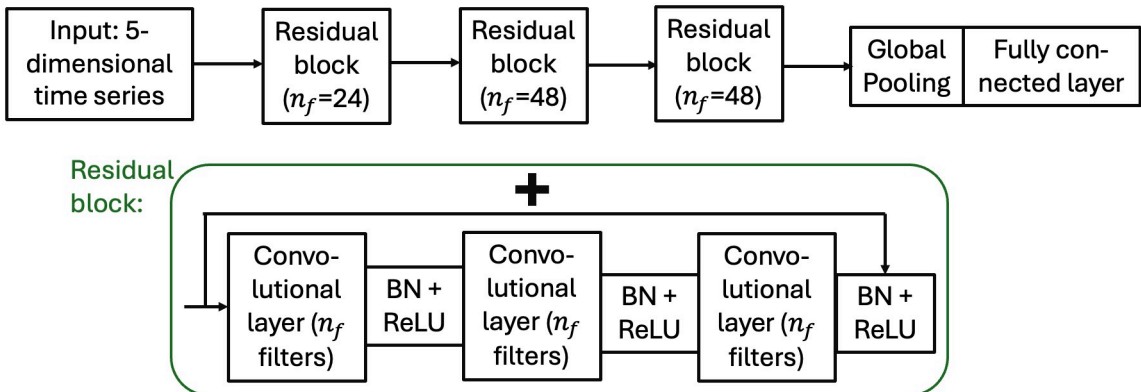

**Fig 5. Architecture of the residual neural network, based on the architecture from [17].** Upper part shows the overall structure and the lower part shows the architecture of the residual blocks. BN + ReLU stands for batch normalization followed by ReLU activation.

*DenseNet.* The DenseNet architecture is inspired by the skip connections in ResNet and introduces a key pattern known as "dense connectivity." Unlike ResNet, which sums the input and output at each shortcut connection, DenseNet concatenates the output feature maps from any given layer directly to subsequent layers. This approach ensures that each layer receives a "collective knowledge" from all preceding layers, enhancing feature propagation and reuse [19]. Overall, the DenseNet architecture is divided into three levels from simple to sophisticated: dense module, dense block, and dense architecture.

Dense modules consist of two convolutional layers (see Fig 6(a)). In the context of this paper, these layers are adapted to one-dimensional (1D) operations to handle time series data, rather than the two-dimensional (2D) operations typically used for image processing. The kernel sizes for convolutional layers were 1 and 3 respectively, and they were operated with 16 filters. Each dense layer employs a residual connection, where the outputs are added to the inputs, facilitating the flow of gradients during training. Hence, after every dense module, the number of features increases by 16, enriching the information collected by the model (see Fig 5(a)).

Each dense block (see Fig 6(b)) is built from a positional encoding layer [20] to incorporate the sequence order of the data, followed by four dense layers, and a 1D max pooling layer to reduce the temporal dimensionality. The dense connectivity principle is fully applied here, meaning the output from each layer is concatenated to every subsequent layer within the block, exponentially increasing the feature maps passed along the network. After every dense block, the number of feature maps increases by 64 and the temporal length is halved.

The overall network structure, the dense architecture (see Fig 6(c)), starts with a single convolutional layer that transforms the input time series, which is represented by an $l$-by-5 matrix with $l$ being the series length, into an initial feature map of size ($l/2$)-by-24. This map is fed into four sequential dense blocks, each enhancing the feature set before passing it to the next, culminating in a robust feature representation suitable for further analysis or classification tasks. The number of input features for the next dense block always increases by 64 (= 16*4). The last dense block outputs a matrix of size ($l/16$)-by-280. Two 1D convolutional layers were applied to transform the output first to size ($l/16$)-by-32 and then to size ($l/16$)-by-16. Finally, the output ($l/16$)-by-16 is flattened to provide an output vector for the regression task.

*Xception network.* The Xception network in this paper follows [21]. This network type consists of Xception modules (see Fig 7(a)) that include two paths, a 3-depthwise separable convolution and a max pooling path, and are linked by residual connections. This architecture

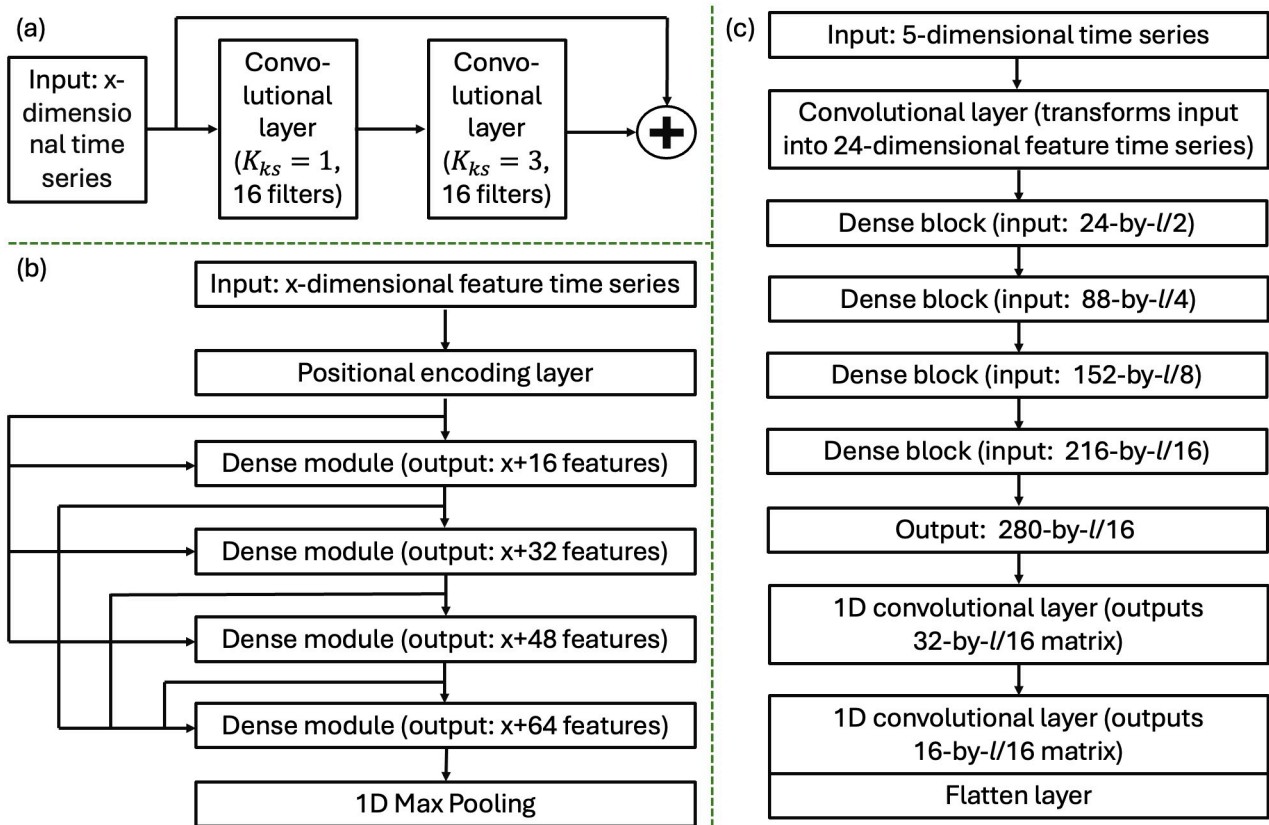

**Fig 6. Architecture and building blocks of DenseNet.** Dense module architecture is shown in (a), dense block architecture in (b), and overall architecture in (c).

enables Xception networks to learn the input with various kernel sizes to obtain both the long and short-term structure in the data [21].

Depthwise separable convolutions are a variant of traditional convolutions that help to reduce complexity while maintaining the performance [22]. They split convolutions into a depthwise convolution and a pointwise convolution. The depthwise convolution applies a single convolutional filter to each input channel independently. In other words, it performs spatial convolution separately for each channel. The pointwise convolution combines the outputs of the three depthwise convolutions across all channels.

Similarly to ResNet, a Xception network incorporates residual connections to mitigate the vanishing gradient problem. However, while Resnet uses standard convolution with high computation complexity, the Xception network combines residual connections, which enables easy gradient flow during training, and depthwise separable convolutions, which enhances efficiency and performance (see [22] for details).

Fig 7(b) illustrates the complete Xception network architecture, which comprises multiple Xception modules stacked together and residual connection links. The one-dimensional adaptive averaging pooling layer aggregates features, retaining information while reducing dimensionality. Similarly to the other tested network types, the ReLU activation function and batch normalization are used after convolution layers to introduce nonlinearity and normalize feature maps, thus enhancing the network's ability to learn complex patterns. In the tests $n_f = \{8, 16\}$ were used and the output vector had size 16 or 32 respectively (variable $c_{out}$).

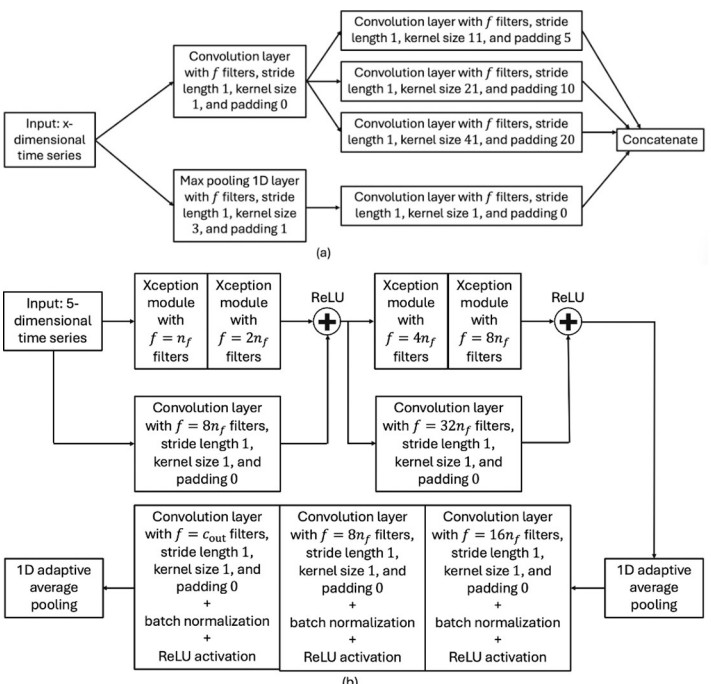

**Fig 7. Architecture and building blocks of XceptionNet.** Architecture of an Xception module is shown in (a), and the Xception network architecture is illustrated in (b).

## Results

### Correlation analysis

In [8] speed, speed change, step duration, vertical displacement, and heart rate were used as input features. This choice was validated by the consider-only-one and leave-one-out approaches. The correlation analysis in [8] revealed that the target feature oxygen consumption ($\dot{V}O_2$) had highest correlations to input features speed, heart rate and speed change, while it was only weakly negatively correlated to vertical oscillation and step duration. Still, adding the latter two as input features to the LSTM model in [8] improved the accuracy of $\dot{V}O_2$ estimations somewhat. In order to being able to compare the data from [8] with the data used in this paper, the correlation analysis was repeated for this paper.

However, one shortcoming of the analysis in [8] was that only Pearson correlation coefficients were computed, which indicate the strength of a linear relationship between two features but do not provide information on potential nonlinear statistical relationships. Therefore, for this paper additionally Spearman correlation coefficients were calculated. Spearman correlation coefficients describe how well a monotonic function describes the relationship between two features. These coefficients do not rely on normality of the data and are robust to outliers due to being a nonparametric measure of rank correlation.

The correlation coefficients for raw data are displayed in Fig 8. The Pearson correlation coefficients (Fig 8(a)) show that heart rate, speed, and speed change have the highest linear correlation with oxygen consumption. It is interesting to note that the correlation coefficients for speed, vertical oscillation and step duration are approximately the same as in [8] (differences at most 0.05) but that the coefficients for heart rate and speed change were approximately 0.2 and 0.3 higher than in [8].

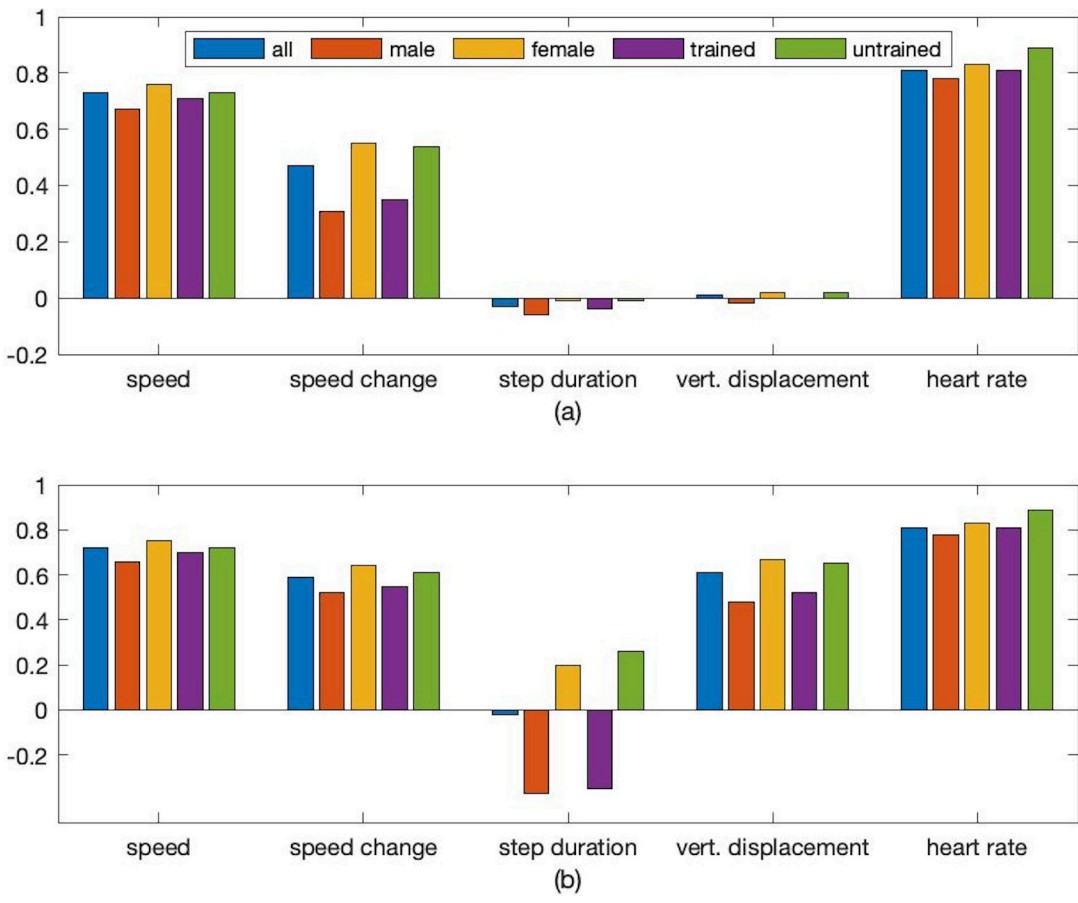

**Fig 8. Correlation coefficients between oxygen consumption (target feature) and five input features.** Input features included speed, speed change, step duration, vertical displacement, and heart rate. Pearson and Spearman correlation coefficients are shown in (a) and (b) respectively.

Looking at the Spearman correlation coefficients (Fig 8(b)) it can be noted that correlations between oxygen consumption and heart rate and speed respectively are (almost) the same as the Pearson coefficients, which could be expected due to a linear function being a monotonic function. For speed change the Spearman coefficient is 0.12 larger, supporting the assumption that the relationship between speed change and oxygen consumption is only approximately linear. The most interesting finding is, however, that vertical oscillation and oxygen consumption are strongly correlated (0.61). Together with the low Pearson coefficient (0.01) this indicates that the relationship between these two features is highly nonlinear and explains why adding it as predictor to the LSTM model in [8] improved the estimation accuracy. On the contrary, the Spearman coefficient of step duration and oxygen consumption is, similarly to the Pearson coefficient, approximately zero.

However, comparing the Pearson and Spearman correlations for data from only male with those for data from only female participants reveals a potential explanation why step duration is a useful input feature for estimating oxygen consumption. It is negatively correlated to oxygen consumption for male (-0.37) but positively correlated for female (0.2). This indicates that using step duration as an input feature could improve accuracy if gender is taken into account. Since the analysed dataset was gender balanced this relationship was hidden when considering the whole dataset. Other features that showed noticeable differences in correlation with oxygen

consumption between male and female participants were speed change (both Pearson and Spearman coefficients) and vertical oscillation (only Spearman coefficient).

Interestingly enough, similar correlation values and differences as for male vs. female participants were found for trained vs. untrained participants (fitness level set to 1 vs. 0), even so both the trained and the untrained group had a male-to-female ratio of one. Furthermore, it was noticed that heart rate and oxygen consumption showed very high Pearson and Spearman coefficients for untrained participants, indicating an almost linear relationship between heart rate and oxygen consumption.

Overall the analysis suggested that the data for this paper are comparable to the data used in [8]. Hence, it was hypothesised that the LSTM model from [8] should yield accurate $\dot{V}O_2$ with the new dataset.

### Intra-subject estimations

For comparing the performance of the Modified LSTM model with the performance of the LSTM model [8] the root mean square error for the $\dot{V}O_2$ estimations over all 16 subjects was calculated. While the average RMSE for the LSTM model was 3.3459 ml×min$^{-1}$×kg$^{-1}$ (standard deviation: 2.3568 ml×min$^{-1}$×kg$^{-1}$), it was 0.6019 ml×min$^{-1}$×kg$^{-1}$ (standard deviation: 0.3076 ml×min$^{-1}$×kg$^{-1}$) for the Modified LSTM model. Fig 9(a) and 9(b) show the Bland-Altman analysis for both model configurations. For the LSTM model the estimation bias was -0.4356 ml×min$^{-1}$×kg$^{-1}$, which is approximately 0.8712% of peak $\dot{V}O_2$, while the bias of the Modified LSTM model was only -0.0078 ml×min$^{-1}$×kg$^{-1}$, which is approximately 0.0156% of peak $\dot{V}O_2$. The validity of estimated oxygen consumption expressed by 95% limits of agreement were 8.2292 ml×min$^{-1}$×kg$^{-1}$ (approximately 16.4584% of peak $\dot{V}O_2$) for *LSTM model* and 1.5470 ml×min$^{-1}$×kg$^{-1}$ (approximately 3.0939% of peak $\dot{V}O_2$) for Modified LSTM model. This shows that by some simple improvements to the architecture of the LSTM model its accuracy for intra-subject $\dot{V}O_2$ estimation can be improved considerably.

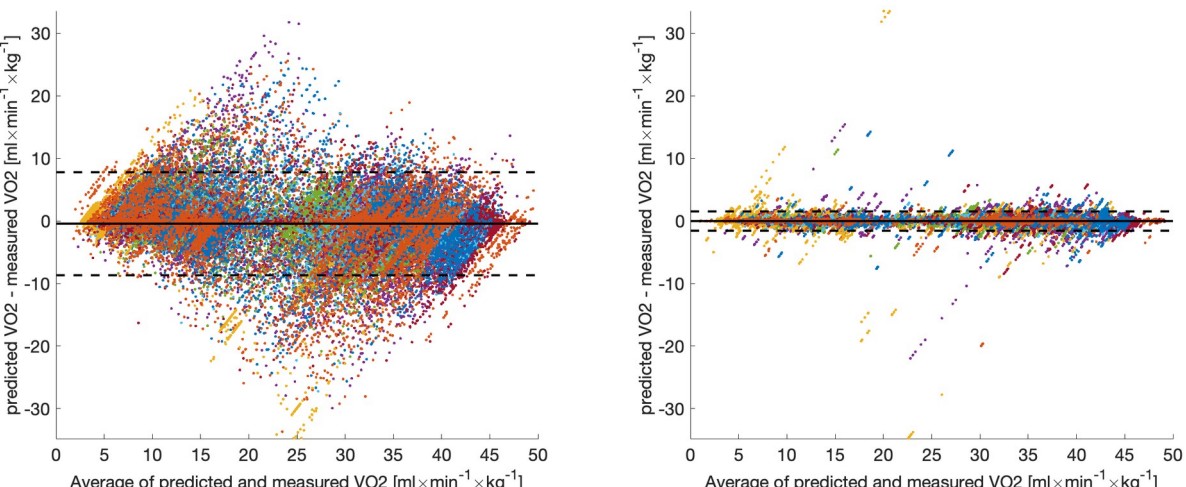

**Fig 9. Bland–Altman analysis of the estimated and directly measured oxygen consumption.** Figure on the left shows results for the LSTM model from [8] and figure on the right shows results for the modified LSTM model with data from all 16 subjects. Dashed horizontal lines represent the 95% limits of agreement and solid lines represent estimation biases. Each color represents data from a unique participant in the test set. For better comparability the y-axes are equally scaled.

**Table 2. Results for a selection of neural network configurations for inter-subject estimation of oxygen consumption.**

| No. | Network type | RMSE (mean) | RMSE (std) | part. spec. | seq. len. | model configurations |
|---|---|---|---|---|---|---|
| 1 | XceptionNet | 2.4295 | 0.2128 | TRUE | 200 | $n_f = 16$, $c_{out} = 16$ |
| 2 | XceptionNet | 2.6109 | 0.3832 | FALSE | 200 | $n_f = 16$, $c_{out} = 16$ |
| 3 | XceptionNet | 2.6262 | 0.3616 | TRUE | 200 | $n_f = 16$, $c_{out} = 32$ |
| 4 | XceptionNet | 2.7328 | 0.5506 | FALSE | 200 | $n_f = 16$, $c_{out} = 32$ |
| 5 | XceptionNet | 2.7533 | 0.4526 | TRUE | 200 | $n_f = 8$, $c_{out} = 32$ |
| 6 | XceptionNet | 3.0478 | 0.8670 | TRUE | 200 | $n_f = 8$, $c_{out} = 16$ |
| 7 | XceptionNet | 3.0600 | 0.6613 | FALSE | 200 | $n_f = 8$, $c_{out} = 32$ |
| 8 | ResNet | 3.0751 | 0.5092 | TRUE | 200 | $n_f = 24$, $c_{out} = 16$, $kss = [3, 3, 3]$ |
| 9 | RNN | 3.0865 | 0.1078 | TRUE | 200 | one-directional GRU |
| 10 | ResNet | 3.1489 | 0.4169 | FALSE | 200 | $n_f = 24$, $c_{out} = 16$, $kss = [7, 5, 3]$ |
| 11 | CNN 16 | 3.3202 | 0.5633 | FALSE | 200 | |
| 12 | RNN | 3.3328 | 0.7950 | FALSE | 200 | one-directional GRU |
| : | | | | | | |
| 14 | CNN 16 | 3.3694 | 0.3840 | TRUE | 200 | |
| : | | | | | | |
| 20 | XceptionNet | 3.5048 | 0.1497 | FALSE | 50 | $n_f = 16$, $c_{out} = 16$ |
| : | | | | | | |
| 24 | CNN | 3.6048 | 0.8707 | FALSE | 200 | |
| : | | | | | | |
| 32 | DenseNet | 3.9190 | 0.9900 | FALSE | 200 | |
| : | | | | | | |
| 37 | CNN | 4.4999 | 1.2157 | TRUE | 200 | |
| : | | | | | | |
| 58 | DenseNet | 7.9411 | 9.1471 | TRUE | 200 | |

Column *Network type* shows the type of network, column *part. spec.* is TRUE if participant- specific features were used in the network and FALSE otherwise. Columns *seq. len.* and *model configurations* yield the length of input sequences and information on configuration hyperparameters respectively. Columns *RMSE (mean)* and *RMSE (std)* contain the average root mean square errors and the corresponding standard deviations. Networks are ordered in ascending order with respect to their average RMSE. The first column contains rankings of the shown network configurations. Results for all 60 tested configurations can be found from the S1 Table.

## Inter-subject estimations

Table 2 lists the results of a selection of network configurations for inter-subject oxygen consumption estimation, sorted with respect to the average root-mean square error of the oxygen consumption estimations over all cross-validation rounds (column *RMSE (mean)*). The results from all network configurations can be found in the supporting information (S1 Table).

For all configurations the average RMSE of test data and the corresponding standard deviations are shown. All five neural network types were tested with input sequences of 50 and 200 steps (column *seq. length*), meaning that the input tensors were of dimensions 5-by-50 and 5-by-200 respectively. In addition, the impact of using participant-specific features such as age, sex, body mass index and fitness level by enabling the optional MLP network for categorical variables was tested (variable *part. spec.* set to TRUE; if FALSE then participant-specific features were not used).

## Discussion

The aim of [8] was to demonstrate that LSTM networks can successfully estimate oxygen consumption when the network is trained with data from the same individual (intra-subject

estimation), and analyse which input parameters would yield the most accurate estimates. However, no attempts at optimizing the network structure or testing alternative architectures were made in [8]. The results in Subsection *Intra-subject estimations* suggest that by including some early exit strategies into the LSTM network, reducing the number of hidden layers in the LSTM layer, and introducing an adaptive learning rate the estimation accuracy can be improved tremendously. Using these modifications, the average root-mean square error for oxygen consumption estimates was reduced by approximately 82% compared to the RMSE of the LSTM network from [8]. The corresponding standard deviation was reduced by approximately 87%. Assuming a peak $\dot{V}O_2$ of 50 ml×min$^{-1}$×kg$^{-1}$, the average RMSE of the Modified LSTM model was 1.2038% of peak $\dot{V}O_2$, which suggests that the setup consisting of INS/GPS datalogger, heart rate monitor, and Modified LSTM model, may yield accurate enough $\dot{V}O_2$ estimates for some monitoring applications, but it remains to be determined whether this approach can detect small, long-term changes in $\dot{V}O_2$, e.g. as a result of training.

For inter-subject $\dot{V}O_2$ estimation preliminary tests with the simple network structure from intra-subject $\dot{V}O_2$ estimation yielded poor accuracy. Thus, more sophisticated network architectures were tested in Subsection *Inter-subject estimations*. The most accurate $\dot{V}O_2$ estimations were achieved by XceptionNet using sequences of 200 steps, participant-specific input features, $n_f$ = 16, and 16 neurons in the output vector. The achieved RMSE of 2.4295 ml×min$^{-1}$×kg$^{-1}$ (standard deviation of 0.2128 ml×min$^{-1}$×kg$^{-1}$) is lower than the LSTM model from [8] used for intra-subject estimations. Even without the use of participant-specific features the RMSE of this XceptionNet configuration increased only to 2.6109 ml×min$^{-1}$×kg$^{-1}$ (+7.47%; standard deviation of 0.3832 ml×min$^{-1}$×kg$^{-1}$). Even more promising is the fact that also using $n_f$ = 8 and/or 32 neurons in the output vector did not result in considerably worse performances. The seven best configurations are all XceptionNet configurations, which suggest that XceptionNet is the most promising network type for inter-subject oxygen consumption estimation.

The best non-XceptionNet configurations are a ResNet configuration in eight and a RNN configuration in ninth place. Their *RMSE test* are 26.58% respectively 27.05% larger than that of the best XceptionNet. For CNN, the best accuracy was achieved with 16 neurons in the last layer before the regression layer, input sequences of 200 steps, and without using participant-specific features (11th best configuration). The best CNN with 32 neurons in the last layer before the regression layer (24th best configuration) yielded a 8.57% higher *RMSE test* than the best CNN 16. The worst network type for $\dot{V}O_2$ estimation is, based on this study, DenseNet, with the best configuration only yielding the 32nd best *RMSE (mean)* for the test data.

Overall, XceptionNet and to some extend ResNet yielded lower RMSEs than the remaining neural network architectures for most of their configurations. A common feature of XceptionNet and ResNet is the use of residual connections. Thus, it could be assumed that the use of these connections and especially combinations of them in XceptionNet is the main reason for these networks low RMSEs. This hypothesis is supported by several studies. For example, [18] and [22] demonstrated the effectiveness of residual connections for mitigating the vanishing gradient problem, thereby enhancing the training stage of deep networks. In addition, [23] demonstrated that residual networks significantly improve the performance of convolutional neural networks for sequence modeling tasks on sequential data. [24] proposed the SAR-UNet model, which integrates residual connections and depthwise separable convolutions. The model has shown substantial improvement in training efficiency and accuracy for forecasting tasks, which involve predicting future states based on time-series data.

When it comes to length of input sequences, 200 steps seems to be a better choice than 50 steps, which is in contrast to the results in [8]. The best network configuration using input

sequences of 50 steps is an XceptionNet with a *RMSE test* of 3.5048 ml×min$^{-1}$×kg$^{-1}$, which is 44.26% larger than the best overall network. The reason for the discrepancy between results from [8] and this paper is most likely that the data available for training was significantly smaller in [8], which only considered intra-subject $\dot{V}O_2$ estimation and participants walking and running at four different speeds.

Based on the study, no clear conclusion on the use of participant-specific features can be drawn. For example, for XceptionNet configurations omitting these features but using otherwise the same configuration resulted in six cases in 4.06% to 43.86% higher *RMSE test*, but in two cases the *RMSE test* was 37.58% to 39.43% higher when using participant-specific features than without them. For RNN configurations, however, using these features yielded in ten of twelve cases 2.32% to 15.93% higher *RMSE test* (for two cases it reduced the *RMSE test* by 2.51% to 7.39%).

## Conclusions

This paper had two aims. The first aim was to find techniques that would significantly increase the accuracy of intra-subject oxygen consumption estimations using the LSTM neural network architecture proposed in [8]. This aim was achieved by including some early exit strategies into the LSTM network and modifying some of the network hyperparameters. The changes resulted in an average root-mean square error reduction from 3.3459 ml×min$^{-1}$×kg$^{-1}$ to 0.6019 ml×min$^{-1}$×kg$^{-1}$.

These promising results encouraged us to investigate a more demanding task, namely developing neural networks that are able to provide accurate oxygen consumption estimates even for inter-subject estimations (second aim). Preliminary attempts with the simple network structures that worked well for intra-subject estimation yielded poor accuracy. Thus, our research focused then on studying five different state-of-the-art neural network architectures with various configurations. The results, especially those of the XceptionNets, suggest that it is indeed possible to accurately estimate the oxygen consumption of an individual from motion and heart rate data using neural networks, even when the data on which the network is trained were collected from other individuals, in other environmental conditions. This second result is of higher importance as it has more relevance for real-world applications.

The achieved accuracy of XceptionNet was at least comparable with previously published methods, suggesting that the presented method could be used by athletes and running enthusiasts to monitor their oxygen consumption over time to detect changes in their movement economy due to training, rehabilitation, etc. However, the accuracy of XceptionNet for oxygen consumption estimation should be verified also in other movements, such as skiing or swimming, and its ability to track small long-term changes in oxygen consumption over the course of several months or even years should be investigated.

In the current setup, oxygen consumption was determined post hoc, i.e. after motion and heart rate data were obtained. In the future, the XceptionNet will be embedded in the datalogger used in this study to provide estimates for oxygen consumption during exercising, once sufficiently long input sequences of motion and heart rate data are available.

One limitation of the training procedure was that each participant walked/ran at the same absolute speeds, but the anaerobic threshold (AT) speed can differ significantly. For example, in [25] the average velocity at AT (vAT) varied from 4.17 m/s to 5.36 m/s for male and from 4.17 m/s to 4.81 m/s for female elite distance runners. Therefore, it is reasonable to assume a lower average vAT and larger variation in vAT for the participants studied in this paper as the group included participants that identified as recreational runners and some that did not. If training data were collected from subjects with high vAT but $\dot{V}O_2$ should be estimated for a

subject with low vAT the estimation accuracy could suffer. Future research could examine this idea using data from individualised vAT-based speeds. Moreover, future work could attempt to estimate oxygen consumption in very low and very high intensity exercises, across multiple testing days, as well as develop male/female specific models with larger datasets. Finally, strategies for updating the estimation models with new unlabeled data should be developed, as the model will often be required to make predictions for a new participant from whom data have not yet been collected. Once data from the new user are added, the model could be fine-tuned to yield more accurate $\dot{V}O_2$ estimates in the future.

Another limitation is that due to the use of input sequences of 50 to 200 steps the proposed setup would be unable to quickly react to changing oxygen consumption due to prompt and significant changes in gait speed. Such highly dynamic settings could be found, for example, in football or sprint disciplines. Developing methods for oxygen consumption estimation in such settings would require different experimental settings including a spirometer that samples $\dot{V}O_2$ reference measurements at a considerably higher rate than the spirometer used in the experiments for this paper.

## Supporting information

**S1 Table. The table contains results for all 60 tested neural network configurations for inter-subject estimation of oxygen consumption and uses the same structure as Table 2.** Networks are ordered in ascending order with respect to their average RMSE. (PDF)

## Acknowledgments

The authors thank Prof. Taija Juutinen for helping with the application for ethical approval. The authors also thank Lauri Orava and Mikko Salonen for their involvement in the data collection.

## Author Contributions

**Conceptualization:** Philipp Müller, Anton Rauhameri.

**Data curation:** Neil J. Cronin.

**Formal analysis:** Philipp Müller, Khoa Pham-Dinh, Huy Trinh.

**Funding acquisition:** Neil J. Cronin.

**Investigation:** Philipp Müller, Khoa Pham-Dinh, Huy Trinh.

**Methodology:** Philipp Müller, Khoa Pham-Dinh.

**Project administration:** Philipp Müller, Neil J. Cronin.

**Software:** Philipp Müller, Khoa Pham-Dinh, Huy Trinh.

**Supervision:** Philipp Müller, Anton Rauhameri, Neil J. Cronin.

**Validation:** Philipp Müller, Khoa Pham-Dinh, Huy Trinh.

**Visualization:** Philipp Müller, Khoa Pham-Dinh.

**Writing – original draft:** Philipp Müller, Khoa Pham-Dinh.

**Writing – review & editing:** Philipp Müller, Khoa Pham-Dinh, Huy Trinh, Anton Rauhameri, Neil J. Cronin.

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
