## [Decision Letter · Decision Letter 0]

18 Jun 2024

PONE-D-24-14729Estimating intra-subject and inter-subject oxygen consumption in outdoor human gait using multiple neural network approachesPLOS ONE

Dear Dr.  Müller,

Thank you for submitting your manuscript to PLOS ONE. After careful consideration, we feel that it has merit but does not fully meet PLOS ONE’s publication criteria as it currently stands. Therefore, we invite you to submit a revised version of the manuscript that addresses the points raised during the review process.

We look forward to receiving your revised manuscript.

Kind regards,

Aamna AlShehhi, PhD

Academic Editor

PLOS ONE

Journal Requirements:

5. Please ensure that you refer to Figure 1 in your text as, if accepted, production will need this reference to link the reader to the figure.

6. We note you have included a table to which you do not refer in the text of your manuscript. Please ensure that you refer to Table 2 and 3 in your text; if accepted, production will need this reference to link the reader to the Table.

Reviewers' comments:

Reviewer's Responses to Questions

**Comments to the Author**

1. Is the manuscript technically sound, and do the data support the conclusions?

Reviewer #1: Partly

Reviewer #2: Yes

2. Has the statistical analysis been performed appropriately and rigorously? 

Reviewer #1: Yes

Reviewer #2: Yes

3. Have the authors made all data underlying the findings in their manuscript fully available?

Reviewer #1: No

Reviewer #2: Yes

4. Is the manuscript presented in an intelligible fashion and written in standard English?

Reviewer #1: Yes

Reviewer #2: Yes

5. Review Comments to the Author

Reviewer #1: The authors have studied the problem of estimating oxygen consumption in outdoor human gait using different deep neural networks. The problem studied is interesting, and can be helpful in analyzing organs in physical activities.

Generally, I would say that the contribution is good. However, it lacks a good presentation. It seems that the paper has been written very quickly, lacking enough attention to details. Thus, I recommend a careful rewrite, especially in the section of networks’ architectures. I also suggest the following modifications:

1- Title of the paper: the phrase “using multiple neural network approaches” sounds strange. It is better to replace it with “using deep neural networks”

2- When abbreviating a word, it is better to start the word you would like to abbreviate with a capital letter, and then put the abbreviation in parentheses. For example: Hearth Rate (HR)

3- When introducing a new concept for the first time, then it is emphasized (italic) and later use of this concept shouldn’t be italic. You have made “LSTM” italic in several places between the lines 168-197. Please correct it

4- The reference in Line 181 is missed

5- Please arrange the features of the test subjects, described in Lines 218-213, in a well-organized table.

6- Line 247 has an open parenthesis without closing

7- Please re-illustrate all figures in the paper and improve their quality

8- “Xception” should always be used with a capital “X”. Please correct it in all places

9- Please recheck the usages of emphasized (italic) words in the whole text

10- Please make the implementations available in a public repository to check validity of the results

Reviewer #2: I read with eager interest the paper “Estimating intra-subject and inter-subject oxygen consumption in outdoor human gait using multiple neural network approaches.” My impression of the manuscript is extremely positive.

This is a well-conceived, well-executed, and well-written study. However, I believe the manuscript could be further improved to enhance the clarity of some messages, terminology, and structure. I hope the authors will consider the following comments as constructive feedback intended to benefit both the authors and readers.

In some cases, the authors could better highlight the positive contributions of this manuscript to the literature. In other cases, certain passages need to be clarified for the readers. Additionally, readers might appreciate it if the conclusion is framed to be relevant to the readership of PLOS ONE, as it is currently quite technical.

Strengths of the manuscript include providing valuable insights into the difference between inter and intra-subject estimation, making the dataset fully available, building on previous open-access work, and conducting the study outdoors rather than on treadmills, which is a noteworthy point to highlight. The authors also address the limitation regarding the presence of ventilatory thresholds, which can significantly affect VO2 predictions.

The authors state in the abstract that “[This technology] could be embedded in portable devices for real-time estimation of oxygen consumption during walking and running.” However, the authors did not assess the capability of this system as a real-time model, making this statement misleading. The authors did not report computational time or the energy resources needed to run the neural network on a portable device. Implementing these neural networks in production involves several challenges beyond input feature measurement, including anthropometric data reception. Therefore, it is premature to conclude that this neural network can be embedded in portable devices, as this work primarily focuses on accuracy assessment, not the performance of the models on real-time embedded devices.

On this note, a limitation of the methodology that was not discussed is that it may not respond promptly to changes in gait speed due to the use of long data windows and the rolling average of VO2. In fact, no time series are reported in the manuscript, which again makes it difficult to assess the potential of this application to work on a real-time basis. Bland-Altman is gold-standard (ish) to evaluate accuracy of measurements, but makes it difficult for the reader to understand how the system behaves when walking speed or running velocity changes in real-time. Again, I do not think authors should place too much emphasis on the real-time application, since time-series were not evaluated and errors in the estimation are not plotted or discussed on the time basis, and the experimental settings have not been designed to assess the performance of the system during transitions and highly dynamic settings. Authors might want to add to the discussion about these points.

The implications for the use of this technology in outdoor and natural settings should be clarified. Can the technology reliably monitor physical activity? What are the main application settings, and can the authors contextualize the results within these applications? Is the accuracy sufficient for the intended applications?

In comparison with the current literature, this manuscript offers an important innovation: the use of multi-head structures in predicting VO2 at the inter-subject level, which is a valuable addition. I congratulate the authors for this brilliant solution. However, it is crucial to clarify how the dataset was used and ensure that data from one subject did not contaminate the dataset used to train/test the neural networks for other subjects. For example, the authors should explicitly detail the normalization process, clearly stating that some subjects were excluded from any part of the training process, including feature normalization. For example: at L164-166, please clarify the type of normalization used (e.g., robust, min-max, or standardization) and whether intra or inter-subject data were used for normalization.

Assessing the correlation coefficient for each variable separately can lead to misleading conclusions due to potential cross-correlation between variables and non-normally distributed data. Alternative methodologies for studying feature importance should be considered. The choice of statistical approach with deep learning models should be justified.

Please clearly specify the dimension of the input tensor (e.g., [5x50]?) at L161. This information, though mentioned later, would be beneficial at this stage.

In general, to avoid confusion, it is advisable to separate the Results and Discussion sections.

In the abstract, it is suggested to define XceptionNet or use the structure/type of neural network and report numerical results if possible.

Throughout the manuscript, there seems to be confusion between the terms "parameters" and "variables," and then input features. In modeling, parameters are constant values, while variables describe the system's evolution. Features are those used as input into deep learning and machine learning models. This inconsistency appears in the abstract and throughout the manuscript, including the methods section where the term "breathing variables" is used. It is important to maintain consistent terminology.

Minor comments include:

Consider a slight modification to the title for brevity: “Estimating intra and inter-subject oxygen consumption in outdoor human gait using multiple neural network approaches.”

At L176, please indicate if the test accuracy/loss was checked to determine the number of epochs.

A citation is missing at L181.

A parenthesis appears to be missing at L247.

Please increase the font size in Figure 8.

Remove the word "only" at L355, as it is arbitrary and this should be a Result section.

There is a typo at L362: “wre” should be corrected.

6. PLOS authors have the option to publish the peer review history of their article (what does this mean?). If published, this will include your full peer review and any attached files.

Reviewer #1: No

Reviewer #2: **Yes: **Andrea Zignoli

---

## [Author Response · Author response to Decision Letter 0]

15 Aug 2024

Dear Dr. AlShehhi,

Please find below the reviewers’ comments and our replies. In the updated manuscript modified text has been highlighted in blue.

Sincerely,

Philipp Müller

Reviewer #1

The authors have studied the problem of estimating oxygen consumption in outdoor human gait using different deep neural networks. The problem studied is interesting, and can be helpful in analyzing organs in physical activities.

Generally, I would say that the contribution is good. However, it lacks a good presentation. It seems that the paper has been written very quickly, lacking enough attention to details. Thus, I recommend a careful rewrite, especially in the section of networks’ architectures. I also suggest the following modifications:

Reply by the authors: Thank you for your review and suggestions. We carefully revised our manuscript and hope it is now to your satisfaction. Please find our answers to your specific concerns in the text below.

1- Title of the paper: the phrase “using multiple neural network approaches” sounds strange. It is better to replace it with “using deep neural networks”

Reply: We discussed your suggestion amongst the authors. Some of us felt that “deep” would not correctly address the content of our manuscript. The reason is that there is no definite answer on how many hidden layers are needed to call a neural network deep. Some sources argue that two hidden layers suffice, others call only networks with dozens of layers deep. Therefore, we retained the original title.

2- When abbreviating a word, it is better to start the word you would like to abbreviate with a capital letter, and then put the abbreviation in parentheses. For example: Hearth Rate (HR)

Reply: We fixed it for HR and INS. We also replaced IMU by inertial measurement unit as it used only once in our manuscript.

3- When introducing a new concept for the first time, then it is emphasized (italic) and later use of this concept shouldn’t be italic. You have made “LSTM” italic in several places between the lines 168-197. Please correct it

Reply: Our intention was to highlight that it is the specific LSTM model from Davidson et al. (2023), but we now removed italic from all but its first appearance. The same change was applied to Modified LSTM model, which is the model proposed in our manuscript for intra-subject estimation.

4- The reference in Line 181 is missed

Reply: Thank you for spotting this missing reference. We now added it.

5- Please arrange the features of the test subjects, described in Lines 218-213, in a well-organized table.

Reply: Please find the description of the four categorical features and their categories in the newly added Table 1.

6- Line 247 has an open parenthesis without closing

Reply: Thank you for spotting the missing parenthesis. We now added it.

7- Please re-illustrate all figures in the paper and improve their quality

Reply: All figures have been updated. Figures 1 to 7 have been heavily overhauled to fulfill your wish for more details on the architectures of the used neural networks. We hope that the new figures in the Network architecture for inter-subject estimation section, together we the rewritten parts, help understand the working principles of the different network structures.

8- “Xception” should always be used with a capital “X”. Please correct it in all places

Reply: We fixed this issue.

9- Please recheck the usages of emphasized (italic) words in the whole text

Reply: This comment seems to be related to your third comment. Thus, please refer to our answer there. We use italic also for denoting variables and headers of columns in the table. We opted to not change these to ensure that the reader can quickly notice that these are variables or column names.

10- Please make the implementations available in a public repository to check validity of the results

Reply: We share the data and our implementations via Fairdata, which is the digital preservation service of the Ministry of Education and Culture, Finland. The dataset can be downloaded at https://doi.org/10.23729/a050e440-6f41-498d-8a31-097ff6881544. Please note that during the review process the access is restricted, but that all files will be made openly accessible upon acceptance of our manuscript.

Reviewer #2

I read with eager interest the paper “Estimating intra-subject and inter-subject oxygen consumption in outdoor human gait using multiple neural network approaches.” My impression of the manuscript is extremely positive.

This is a well-conceived, well-executed, and well-written study. However, I believe the manuscript could be further improved to enhance the clarity of some messages, terminology, and structure. I hope the authors will consider the following comments as constructive feedback intended to benefit both the authors and readers.

Reply by the authors: Dear Dr. Zignoli, thank you for your insightful review. We carefully revised our manuscript based on your and the second reviewer’s comments. Please find our answers to your specific concerns in the text below.

In some cases, the authors could better highlight the positive contributions of this manuscript to the literature. In other cases, certain passages need to be clarified for the readers. Additionally, readers might appreciate it if the conclusion is framed to be relevant to the readership of PLOS ONE, as it is currently quite technical.

Reply: We rewrote the conclusion section to make it less technical and better highlight our contributions. Furthermore, we added statements to the abstract and introduction (lines 4-6) that our focus was on oxygen consumption estimation in outdoor environments rather than during exercising on treadmills in laboratory environment.

Strengths of the manuscript include providing valuable insights into the difference between inter and intra-subject estimation, making the dataset fully available, building on previous open-access work, and conducting the study outdoors rather than on treadmills, which is a noteworthy point to highlight. The authors also address the limitation regarding the presence of ventilatory thresholds, which can significantly affect VO2 predictions.

The authors state in the abstract that “[This technology] could be embedded in portable devices for real-time estimation of oxygen consumption during walking and running.” However, the authors did not assess the capability of this system as a real-time model, making this statement misleading. The authors did not report computational time or the energy resources needed to run the neural network on a portable device. Implementing these neural networks in production involves several challenges beyond input feature measurement, including anthropometric data reception. Therefore, it is premature to conclude that this neural network can be embedded in portable devices, as this work primarily focuses on accuracy assessment, not the performance of the models on real-time embedded devices.

Reply: You are right, the statement was misleading. What we meant was that the INS/GPS-based datalogger and the heart rate device measure in real-time. For now, the oxygen consumption is estimated once data collection has been stopped. However, our vision is to further develop our hardware and software to yield VO2 estimates during training based on the datalogger and heart rate measurements and that, based on the presented test results, XceptionNet would be the best choice for computing these estimates. As you correctly commented it will still not be possible to predict oxygen consumption in real-time because an input sequence of several steps will be needed before oxygen consumption can be estimated, but with a short delay (less than a minute). 

We updated the abstract accordingly and modified the conclusion section (lines 521-525) to clarify our vision for future research and development work.

On this note, a limitation of the methodology that was not discussed is that it may not respond promptly to changes in gait speed due to the use of long data windows and the rolling average of VO2. In fact, no time series are reported in the manuscript, which again makes it difficult to assess the potential of this application to work on a real-time basis. Bland-Altman is gold-standard (ish) to evaluate accuracy of measurements, but makes it difficult for the reader to understand how the system behaves when walking speed or running velocity changes in real-time. Again, I do not think authors should place too much emphasis on the real-time application, since time-series were not evaluated and errors in the estimation are not plotted or discussed on the time basis, and the experimental settings have not been designed to assess the performance of the system during transitions and highly dynamic settings. Authors might want to add to the discussion about these points.

Reply: We clarified in the manuscript that in most cases real-time information is not needed. Instead, a report of oxygen consumption during an exercise session is available within minutes after the session. In the future, we will try to embed the estimation method on the datalogger to provide the user VO2 estimates also during exercising, with some small delay (compare our reply to the previous comment).

Changes can be found in the rewritten Conclusion section. For example, we added one paragraph at the end of the section to discuss the limitation that our estimation method might not respond promptly to changing oxygen consumption due to changes in gait speeds in sports such as football. To test this hypothesis would require a different experimental setup including a spirometer with a higher sampling rate to obtain reliable reference measurements.

The implications for the use of this technology in outdoor and natural settings should be clarified. Can the technology reliably monitor physical activity? What are the main application settings, and can the authors contextualize the results within these applications? Is the accuracy sufficient for the intended applications?

Reply: The goal of our approach is to monitor oxygen consumption over time, to observe (for example) changes in movement economy due to training, rehabilitation etc. Therefore, the main application setting would be as a monitoring tool for athletes and running enthusiasts (we added clarification to the first paragraph of the Introduction section (lines 2-9) and the second paragraph of the Conclusions section (lines 502-512)). 

We clarified in Conclusions (third paragraph, lines 513-520) that the accuracy of XceptionNet was at least comparable with the results of previously published methods, indicating that the presented method is accurate enough to be useful as a training tool. However, we acknowledge that we still need to verify the accuracy of the approach in other movements (e.g. skiing, swimming...), as well as the ability to track small long-term changes in oxygen consumption over the course of several months or even years. You will find a short discussion on this issue in the third paragraph of the Conclusions section.

In comparison with the current literature, this manuscript offers an important innovation: the use of multi-head structures in predicting VO2 at the inter-subject level, which is a valuable addition. I congratulate the authors for this brilliant solution. However, it is crucial to clarify how the dataset was used and ensure that data from one subject did not contaminate the dataset used to train/test the neural networks for other subjects. For example, the authors should explicitly detail the normalization process, clearly stating that some subjects were excluded from any part of the training process, including feature normalization. For example: at L164-166, please clarify the type of normalization used (e.g., robust, min-max, or standardization) and whether intra or inter-subject data were used for normalization.

Reply: We clarified in the third paragraph of subsection Network architecture for inter-subject estimations (lines 225-234) the testing procedure for inter-subject estimation. Data from the subject for which oxygen consumption was to be estimated was neither used in training nor validation datasets.

For inter-subject estimation the dataset of a single subject was divided into a training, a validation, and a test set to prevent data leakage.

We regret that this statement was presented in lines 164-166 of our manuscript, since the features were not normalized during dataset preparation. Instead, the normalization was done inside the Python code for intra-subject VO2 estimation. We updated the third paragraph of section LSTM network architecture for intra-subject estimations (lines 195-197) to clarify how those data were standardized. 

We also clarified that standardized oxygen and heart rate data (i.e. data were centered to have zero mean and standard deviation one) were used to synchronize the measurements of the spirometer and heart rate monitor (first paragraph of section Dataset preparation.

Assessing the correlation coefficient for each variable separately can lead to misleading conclusions due to potential cross-correlation between variables and non-normally distributed data. Alternative methodologies for studying feature importance should be considered. The choice of statistical approach with deep learning models should be justified.

Reply: We agree that cross-correlation is one possible method for evaluating feature importance. However, we clarified in Dataset preparation that features were identified in reference [8]. In that paper both consider-only-one and leave-one-out approaches were used to validate the five features. Since our study built on the findings in [8] we decided to use the same features.

We included the correlation analysis to compare the structures of our dataset and the dataset used in [8], and it should be understood as exploratory data analysis (see first paragraph of Correlation analysis). We agree that assessing only Pearson correlation coefficients might lead to wrong conclusions as it is only a measure of the strength of linear relationships. Therefore, unlike [8], we also computed Spearman correlation coefficients as the Spearman rho does not rely on normality of the data and is robust to outliers since it is a non-parametric measure of rank correlation (a clarification has been added to the second paragraph of Correlation analysis, lines 354-356).

Please clearly specify the dimension of the input tensor (e.g., [5x50]?) at L161. This information, though mentioned later, would be beneficial at this stage.

Reply: In the updated manuscript we specify in lines 167-169 that 5-by-50 input tensors were used for intra-subject estimation. For inter-subject estimation the input sensor was either [5x50] or [5x200] (both options were tested with all network architectures). We clarified this in lines 422/423.

In general, to avoid confusion, it is advisable to separate the Results and Discussion sections.

Reply: We split the Results and discussion section into Results section and Discussion section.

In the abstract, it is suggested to define XceptionNet or use the structure/type of neural network and report numerical results if possible.

Reply: We rewrote the abstract to address this comment. It now reports the lowest average estimation error obtained for inter-subject VO2 estimation.

Throughout the manuscript, there seems to be confusion between the terms "parameters" and "variables," and then input features. In modeling, parameters are constant values, while variables describe the system's evolution. Features are those used as input into deep learning and machine learning models. This inconsistency appears in the abstract and throughout the manuscript, including the methods section where the term "breathing variables" is used. It is important to maintain consistent terminology.

Reply: We carefully revised the manuscript to ensure consistent and correct terminology throughout the manuscript.

Minor comments include:

Consider a slight modification to the title for brevity: “Estimating intra and inter-subject oxygen consumption in outdoor human gait using multiple neural network approaches.”

Reply: We updated the title accordingly.

At L176, please indicate if the test accuracy/loss was checked to determine the number of epochs.

Repl

---

## [Decision Letter · Decision Letter 1]

6 Sep 2024

Estimating intra- and inter-subject oxygen consumption in outdoor human gait using multiple neural network approaches

PONE-D-24-14729R1

Dear Dr. Philipp Muller,

We’re pleased to inform you that your manuscript has been judged scientifically suitable for publication and will be formally accepted for publication once it meets all outstanding technical requirements.

Kind regards,

Aamna AlShehhi, PhD

Academic Editor

PLOS ONE

Reviewers' comments:

Reviewer's Responses to Questions

**Comments to the Author**

1. If the authors have adequately addressed your comments raised in a previous round of review and you feel that this manuscript is now acceptable for publication, you may indicate that here to bypass the “Comments to the Author” section, enter your conflict of interest statement in the “Confidential to Editor” section, and submit your "Accept" recommendation.

Reviewer #1: All comments have been addressed

Reviewer #2: All comments have been addressed

2. Is the manuscript technically sound, and do the data support the conclusions?

Reviewer #1: Yes

Reviewer #2: Yes

3. Has the statistical analysis been performed appropriately and rigorously? 

Reviewer #1: Yes

Reviewer #2: Yes

4. Have the authors made all data underlying the findings in their manuscript fully available?

Reviewer #1: Yes

Reviewer #2: Yes

5. Is the manuscript presented in an intelligible fashion and written in standard English?

Reviewer #1: Yes

Reviewer #2: Yes

6. Review Comments to the Author

Reviewer #1: Thanks for addressing all the comments. The revised version of the paper perfectly fits to be published in the PLOSE ONE journal.

Reviewer #2: Thank you to the authors for thoroughly addressing all the comments and for taking the time to revise the manuscript according to the reviewers' recommendations. This work is a valuable contribution to the current literature on the topic.

7. PLOS authors have the option to publish the peer review history of their article (what does this mean?). If published, this will include your full peer review and any attached files.

Reviewer #1: No

Reviewer #2: **Yes: **Andrea Zignoli

---

## [Editor Report · Acceptance letter]

18 Sep 2024

PONE-D-24-14729R1 

PLOS ONE

Dear Dr. Müller, 

I'm pleased to inform you that your manuscript has been deemed suitable for publication in PLOS ONE. Congratulations! Your manuscript is now being handed over to our production team.

Kind regards, 

on behalf of

Dr Aamna AlShehhi 

Academic Editor

PLOS ONE